# Novel Cobalt Complex as an Efficient Catalyst for Converting $CO_2$ into Cyclic Carbonates under Mild Conditions

**Wei Fan [1], Wen-Zhen Wang [1],\*, Li Wang [1], Xin-Gang Jia [1], Lei-Lei Li [1], Tian-Cun Xiao [2] and Peter P. Edwards [2]**

[1]   School of Chemistry and Chemical Engineering, Xi'an Shiyou University, Xi'an 710065, China; vfan111@163.com (W.F.); lwang2018@xsyu.edu.cn (L.W.); jiaxingang76@xsyu.edu.cn (X.-G.J.); lll@xsyu.edu.cn (L.-L.L.)

[2]   Inorganic Chemistry Laboratory, University of Oxford, South Parks Road, Oxford OX1 3QR, UK; xiao.tiancun@chem.ox.ac.uk (T.-C.X.); Peter.edwards@chem.ox.ac.uk (P.P.E.)

\*   Correspondence: wzwang@xsyu.edu.cn; Tel.: +86-133-8921-4744

**Abstract:** Based on the ligand $H_2dpPzda$ (**1**), a novel cobalt complex $[Co(H_2dpPzda)(NCS)_2] \cdot CH_3OH$(**2**) has been synthesized and characterized. The Complex **2** exhibited excellent catalytic performance for converting $CO_2$ into cyclic carbonates under mild conditions. For propylene oxide (PO) and $CO_2$ synthesis of propylene carbonate (PC), the catalytic system showed a remarkable TOF as high as 29,200 $h^{-1}$. The catalytic system also showed broad substrate scope of epoxide. Additionally, the catalyst could be recycled to maintain the integrity of the structure and remained equal to the level of its catalytic activity even after seven catalytic rounds. Additionally, a possible catalytic mechanism was proposed due to the high catalytic activity which might be owing to the synergism of Lewis acidic metal centers and N group.

**Keywords:** supramolecular cobalt complex; $CO_2$ fixation; cyclic carbonates; recyclability

## 1. Introduction

As one of the main gases that produce the greenhouse effect, $CO_2$ destroys the ecological balance of nature and threatens the survival of various organisms. At the same time, its cheapness, abundance and safety make it one of the main research directions of green chemistry [1–3]. Indeed, the synthesis of cyclic carbonates with epoxides is one of the most effective routes to utilize carbon dioxide resources. Cyclic carbonate has quite a lot of application as an intermediate for many chemical syntheses [4,5]. Therefore, there have been many studies on the catalytic synthesis of cyclic carbonates over the past few decades.

The cycloaddition reaction of epoxide and $CO_2$ is generally carried out over various catalysts, including metal complexes [6–8], metal oxides [9], molecular sieves [10], ionic liquids [11,12] and organic catalysts [13–15]. Among these catalysts, metal complexes show superiority due to their easy-to-synthetize, high stability, and abundant spatial structure. However, it has been reported in the past that most of the catalysts for converting $CO_2$ into cyclic carbonates are insufficient, such as harsh reaction conditions, excessive catalyst loading, and the need for solvent to be added. On the other hand, most homogeneous catalysts are non-recycled. Therefore, it is urgently needed to synthesize catalysts with more stable, higher recyclability that can work under milder reaction conditions.

Owing to multiple metal coordination sites and the intermolecular interaction, supramolecular metal catalysts with high catalytic activity and selectivity have emerged as highly promising materials for wide applications, including the cycloaddition reaction of $CO_2$ [16–18]. It has been found that the

nitrogen-containing heterocyclic group possessed by the oligopyrazinediamine ligand has various coordination modes, which can be combined with transition metals under neutral conditions (for example, nickel, manganese, iron, cobalt, etc.) to form a series of metal-organic coordination compounds. Interestingly, a supramolecular catalyst with an oligopyrazinediamine ligand can be synthesized by optimizing the synthesis process [19–22].

Herein, $H_2$dpPzda(1) [22] was used as a ligand, and we have designed and synthesized a novel cobalt complex [Co($H_2$dpPzda)(NCS)$_2$]·$CH_3OH$(2) (Scheme 1). Complex **2**, which was used as a catalyst for the cycloaddition of $CO_2$ and epoxides under mild conditions, exhibited excellent catalytic performance with high selectivity and yield and turnover frequency (TOF). Additionally, the insolubility of Complex **2** in the reactants and products allows the catalyst to be easily recycled by simple centrifugation with almost no loss. Therefore, we propose a possible mechanism that the synergism of Lewis acidic metal centers and N group make it highly catalytically active.

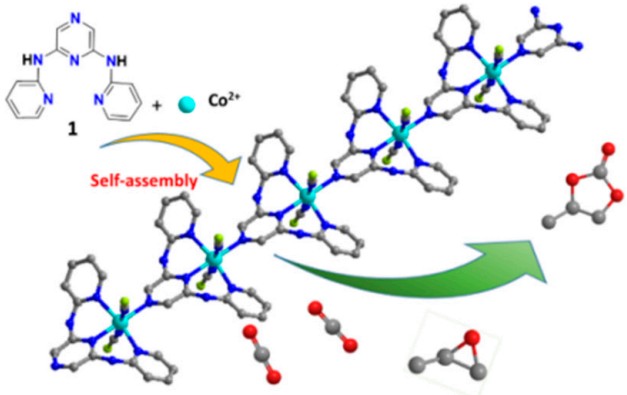

**Scheme 1.** Self-assembly of complex **2** for $CO_2$ conversion (N, blue; C, gray; S, yellow).

## 2. Results and Discussion

### 2.1. Structure and Characterization

Complex **2** was synthesized when Co(NCS)$_2$ reacted with the ligand $H_2$dpPzda in methanol. The experimental conditions, data collection, structural analysis, correction methods and crystal data of the complex related to X-ray diffraction analysis were shown in Table S1. The main bond lengths were listed in Table S2.

Complex **2** was a one-dimensional straight chain polymeric structure of neutral molecular type, and its crystal belonged to the *Aba2* space group of orthorhombic system. The crystal structure diagram of the Complex **2** was shown in Figure 1a. In the Complex **2**, each ligand had four aromatic nitrogen atoms as coordination atoms, the ligand was tetradentate coordination, and the pyrazine ring acted as a bridging ligand to link the small molecular units of the complex into a one-dimensional chain polymerization. Co(II) coordinated with six nitrogen atoms to form an elongated octahedral geometry. The equatorial plane was constructed through four nitrogen atoms, three of which came from one ligand and the fourth nitrogen atom from the pyrazine ring of another ligand. The two nitrogen atoms of the pyrazinyl group and the Co(II) ion were coordinated at the equatorial position. In the axial position, the other two nitrogen atoms were derived from thiocyanate. Adjacent ligands were on the same plane (Co–Co–Co = 180.00°) and the spacing between Co–Co reached 7.14 Å. The bond length of the Co–N ($H_2$dpPzda) bond varied from 2.121 to 2.202, and the bond length of the Co-N (NCS-) bond was 2.059 Å which indicated that the anionic NCS had strong binding ability to Co(II). The N-C-S participating in the coordination of the N-C-S had an angle of 179.30° and was almost straight. For Co(II) ions, the coordination environment could be considered as an elongated square bipyramid (Figure 1b). All of the bond lengths of Co–N fell into the normal ranges. In Complex **2**, the ligand coordinated in a bidentate bridging mode with two ligands in a *cis* conformation. The hydrogen bonded N2-H2···O1

and O1-H1B···S1 between the amine group, and the thiocyanate and the methanol solvent (Table S3) transformed the one-dimensional chain structure into a two-dimensional plane as shown in Figure 1c.

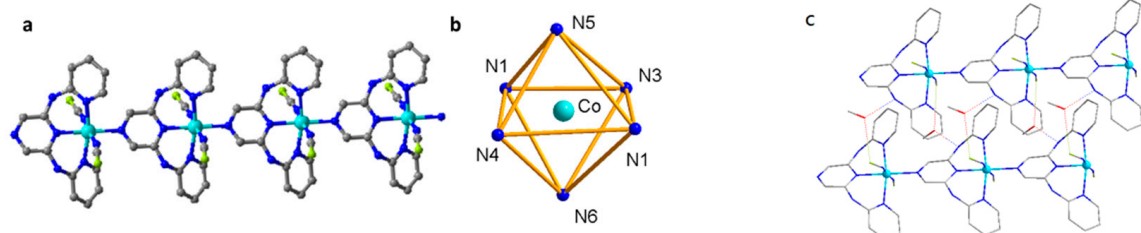

**Figure 1.** (**a**) The coordination environment of complex **2** (hydrogen atoms were omitted for clarity); (**b**) the coordination polyhedron of $Co^{2+}$; and (**c**) 2-D supramolecular structure of complex **2** with H atoms omitted for clarity. H bonds were indicated with dashed lines. (Hydrogen atoms were omitted for clarity. Color scheme: Co atoms, light blue; C atoms, gray; N atoms, blue; S atoms, yellow; O atoms, red).

## 2.2. Catalytic Activity for $CO_2$ Fixation

In this section, we designed a series of experiments to explore the potential catalytic performance of compounds **1** and **2**. The pyridyl and cobalt metals in **2** could give the catalyst a high density of Lewis-basic and Lewis-acid active sites, giving it multiple functionalities, indicating that they have broad prospects in catalytic applications. Thus, we designed a series of experiments to examine their catalytic activities. As shown in Table 1, the catalyst and the co-catalyst had a significant effect on the cycloaddition of $CO_2$ and propylene oxide (substrate). We utilized 0.025 mol% complex **2** in combination with 0.5 mol% TBAB as binary catalysts to obtain solvent-free cyclic carbonates under the conditions of 90 °C, 1MPa, which achieved excellent conversion (89%) and showed high catalytic activity with initial turnover frequency (TOF) of up to 3560 $h^{-1}$ after 1 h reaction (entry 5, Table 1). We compared it with representative catalysts under different conditions so as to evaluate the high catalytic performance of this catalytic system (Table 2). The results show that the catalytic activity of complex **2** is higher than most of the published catalysts, such as $Zn(OPO)_2$, Al-iPOP-2, Co-Salen and so on. Considering that co-catalysts such as ammonium salts were effective in improving the catalytic activity of complexes, different co-catalysts (DMAP and TBAB) were tested with the best performance of TBAB (entries 5 and 12, Table 1). Neither **1**, **2** nor TBAB alone could catalyze the reaction with satisfactory yields (9%, 16% and 25%, respectively), while the conversion was increased to 89% via the co-existence of **2** and TBAB. However, in the absence of co-catalysts, catalysts **1** and **2** still showed a certain catalytic activity which might be due to the synergism of N groups (entries 2–3, Table 1). The conversion was decreased from 89% to 49% with the temperature dropping from 90 °C to 60 °C, though it showed excellent catalytic activity with relatively high initial TOF >1800 $h^{-1}$ (entries 5–8, Table 1). Nonetheless, the catalytic activity was less sensitive to $CO_2$ pressure than temperature. The conversions remained hardly changed when the $CO_2$ pressure was reduced from 15 to 8 bar, but the conversions would drop dramatically when the pressure dropped to 6 bar (Table 1, entries 9–11). Good yields were still obtained at lower TBAB loading (0.25 mol% and 0.1 mol%, respectively), with the initial TOF of up to 3000 $h^{-1}$ and 2480 $h^{-1}$ (entries 13–14, Table 1), respectively. It was noteworthy that the propylene oxide was used without removal of either oxygen or moisture in advance. Besides, air or water (volume ratio = 20%) (entries 15–16, Table 1) had little influence on the reactions in the catalytic process, indicating that complex **2** had better stability in this process. Gratifyingly, the loading of 0.0025 mol% **2** and 0.5 mol% TBAB for the cycloaddition of PO under the conditions of 120 °C, 1MPa after 1 h reaction, brought about a moderate conversion of 73% and a remarkable TOF as high as 29,200 $h^{-1}$ (entry 18, Table 1). This result proved that the reaction was still extremely active with a little catalyst content.

**Table 1.** Cycloaddition of $CO_2$ and propylene oxide (PO) using **1** and **2** [a].

| Entry | Cat | Co-cat | T(°C) | P(bar) | T(h) | Conversions [b] (%) | TOF [c] ($h^{-1}$) |
|-------|-----|--------|-------|--------|------|----------------------|---------------------|
| 1 | **-** | - | 90 | 10 | 1 | - | - |
| 2 | **1** | - | 90 | 10 | 1 | 9 | 360 |
| 3 | **2** | - | 90 | 10 | 1 | 16 | 640 |
| 4 | **-** | TBAB | 90 | 10 | 1 | 25 | - |
| 5 | **2** | TBAB | 90 | 10 | 1 | 89 | 3560 |
| 6 | **2** | TBAB | 80 | 10 | 1 | 79 | 3160 |
| 7 | **2** | TBAB | 70 | 10 | 1 | 63 | 2520 |
| 8 | **2** | TBAB | 60 | 10 | 1 | 45 | 1800 |
| 9 | **2** | TBAB | 90 | 15 | 1 | 95 | 3800 |
| 10 | **2** | TBAB | 90 | 8 | 1 | 84 | 3360 |
| 11 | **2** | TBAB | 90 | 6 | 1 | 56 | 2240 |
| 12 | **2** | DMAP | 90 | 10 | 1 | 71 | 1840 |
| 13 [d] | **2** | TBAB | 90 | 10 | 1 | 75 | 3000 |
| 14 [e] | **2** | TBAB | 90 | 10 | 1 | 62 | 2480 |
| 15 [f] | **2** | TBAB | 90 | 10 | 1 | 86 | 3440 |
| 16 [g] | **2** | TBAB | 90 | 10 | 1 | 83 | 3320 |
| 17 [h] | **-** | TBAB | 120 | 10 | 1 | 19 | - |
| 18 [h] | **2** | TBAB | 120 | 10 | 1 | 73 | 29,200 |

[a] Reaction conditions: 50 mmol PO, 0.025 mol% catalyst, 0.5 mol% co-catalyst. [b] Determined by [1]H NMR analysis. Selectivity of cyclic carbonates were all >99%. [c] TOF = Conversion/$n_{cat}$/h. [d] 0.025 mol% catalyst, 0.25 mol% co-catalyst. [e] 0.025 mol% catalyst, 0.1 mol% co-catalyst. [f] not excluding the air inside the reaction system. [g] 0.2 mL of $H_2O$ was added to the reaction system. [h] Reaction condition: 50 mmol PO, 0.0025 mol% catalyst, 0.5 mol % TBAB.

To further investigate the catalytic performance of complex **2**, different epoxy substrates were tested under mild conditions. As shown in Table 3, all terminal expoxides could be converted into the corresponding cyclic carbonates with high yields and selectivities under the conditions of 90 °C and 10 bar $CO_2$ after 1h reaction. In addition, epichlorohydrin and allyl glycidyl ether exhibited the best catalytic activity with the highest conversions (99%) and TOF (3960 $h^{-1}$).

**Table 2.** Representative homogeneous and heterogeneous cobalt catalysts with high TOF used for the synthesis of cyclic carbonates.

| Cat. | Co-Cat. | Catalyst/Epoxide (Mole Ratio) | P (MPa) | T (°C) | Time (h) | Conversions (%) | TOF ($h^{-1}$) | Ref |
|------|---------|-------------------------------|---------|--------|----------|------------------|-----------------|-----|
| Co-Salen | DMAP (1) | (Epichlorohydrin) 1:200000 | 0.1 | 120 | 3 | 5 | 3333 | [23] |
| Zn(OPO)$_2$ | TBAB (0.9) | (propylene epoxide) 1:40000 | 3 | 120 | 1 | 46 | 18,400 | [24] |
| Al-iPOP-2 | - | (propylene oxide) 1:10000 | 1 | 100 | 4 | 97 | 7600 | [25] |
| Mg-porphyrin | - | (1,2-Epoxyhexane) 1:33333 | 1.7 | 120 | 1 | 36 | 12,000 | [26] |
| Zn-CMP | TBAB (0.9) | (propylene oxide) 1:25000 | 3 | 120 | 1 | 29 | 11,600 | [27] |
| 2 | TBAB | 1:40000 | 1 | 120 | 1 | 73 | 29,200 | This work |

**Table 3.** Cycloaddition of $CO_2$ to different epoxide substrates using complex **2** as a catalyst.

| Entry | Substrate | Product | Conversion (%) | TOF ($h^{-1}$) |
|-------|-----------|---------|----------------|----------------|
| 1 | | | 89 | 3560 |
| 2 | | | 99 | 3960 |
| 3 | | | 98 | 3920 |
| 4 | | | 97 | 3880 |
| 5 | | | 75 | 3000 |
| 6 | | | 99 | 3960 |

Reaction conditions: 10 mmol substrates, 0.025 mol% catalyst, 0.05 mol% co-catalyst, 90 °C, 10 bar $CO_2$, 1 h. Conversions were determined by $^1$H NMR analysis. Selectivity of cyclic carbonates were all >99%.

The recyclability and stability of a catalyst plays an integral role in practical application [24,28,29]. The recycling experiment was carried out by the optimal reaction conditions obtained under the above experiment, i.e., 60 °C and 10 bar of $CO_2$ and 12 h. 1 mol% of the catalyst and propylene oxide were placed in an autoclave and passed through 10 bar of $CO_2$ for 12 h to calculate the yield. Subsequently, the catalyst was recovered by centrifugation, washed and dried, and reused under the same conditions. The IR spectrum of **2** confirmed that the structural integrity of **2** was well maintained after catalysis (Figure 2b). The cycle of the catalytic system was successfully repeated seven times with little change in conversion (Figure 2a).

Considering the mechanism for coupling reaction of epoxides and $CO_2$ reported earlier, and frontier molecular orbital calculation results (Table S4), we tentatively proposed a possible catalyzed cycloaddition mechanism of complex **2** in Scheme 2 [12,30–32]. Firstly, epoxide was activated by the coordination with multiple Lewis acidic metal centers, and intermediate II was formed. Secondly, nucleophilic reagent Br$^-$ of TBAB intermediate III was formed. Finally, $CO_2$, which was polarized by pyridine groups of complex **2** (N-C = OO), was inserted into the M-O epoxide bonds to generate cyclic carbonates and was epoxidized by an internal attack of $CO_2$ on the C-Br bond with re-generation of the co-catalyst. Experimental evidence showed that besides the high density of Lewis acid active sites, the existence of nitrogen-containing functionalized ligands **1**, which was used in combination with TBAB, was used as a catalyst for the model reaction. Either **1** or **2** TBAB alone could also catalyze the reaction with conversions 9% and 16%, respectively. This mechanism denoted that the high catalytic activity might be due to the synergism of Lewis acidic metal centers and N group.

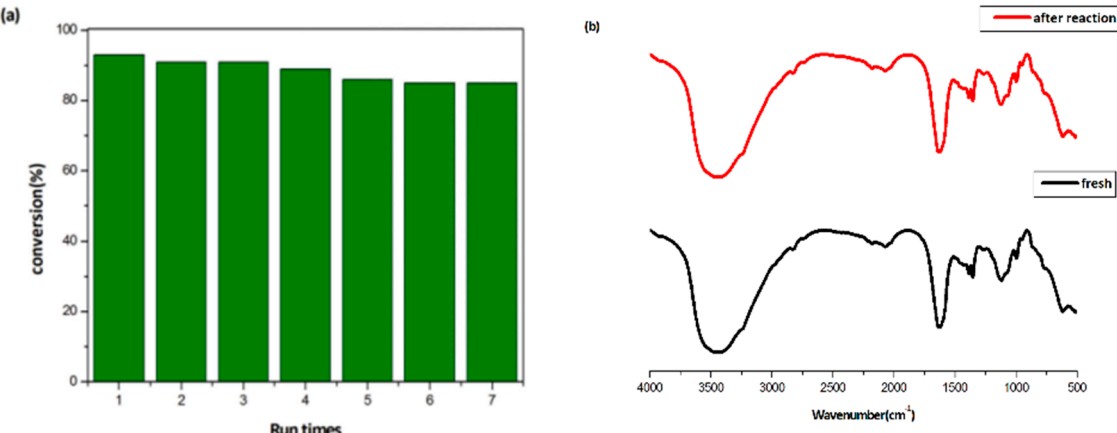

**Figure 2.** (**a**) Recycling experiments of $CO_2$ cycloaddition of propylene oxide. (Reaction conditions: 10 mmol propylene oxide, 1 mol% catalyst, 1 mol% TBAB, r.t., 60 °C, 12 h, 10 bar $CO_2$) and (**b**) IR spectra of complex **2** and its sample after recycled reactions.

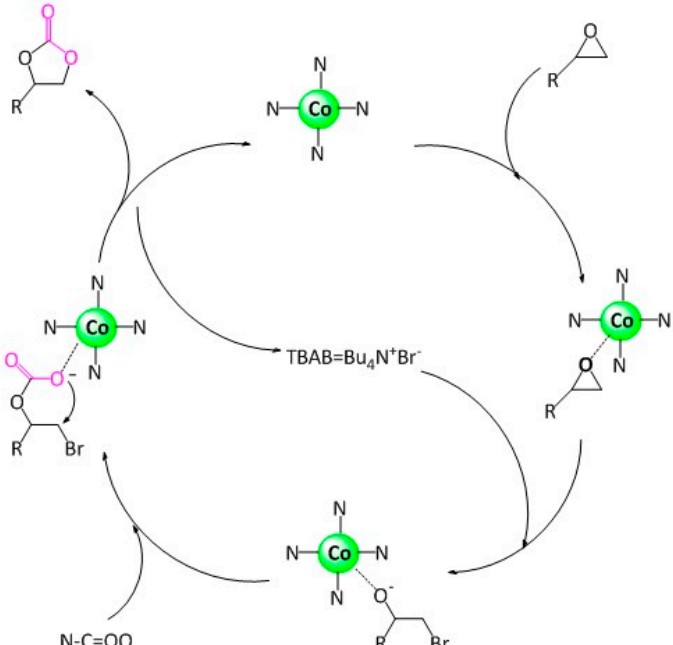

**Scheme 2.** Possible mechanism for cycloaddition of $CO_2$ catalyzed by complex **2**.

## 3. Materials and Methods

### 3.1. Materials and Characterizations

All samples and solvents were of analytical grade and were ready to use. The ligand *N,N'*-Di(pyridin-2-yl)pyrazine-2,6-diamine($H_2$dpPzda) was synthesized following the reported procedure [19]. FT-IR spectra were obtained on a VERTEX-70 Fourier transform infrared spectrometer with a band ranging from 4000 to 400 $cm^{-1}$. The elemental analysis of complex was performed using Elementar Germany's Vario MICRO elemental analyzer. The thermogravimetric analysis data of complex was obtained by using America METTLER TOLEDO thermogravimetric analysis. The UV spectrum data of complex was obtained by using Japan Shimadzu UV-2600 ultraviolet spectrophoto-meter. $^1$H NMR spectra were recorded on a JEOL ECS400M spectrometer with reference to the solvent signals. A single crystal of size $0.30 \times 0.15 \times 0.03$ $mm^3$ was placed on the diffractometer, and the crystal structure was determined by graphite monochromated Mo K*a* (λ = 0.071073 nm) radiation and *ω-2θ* scanning. All calculations were

performed using the SHLXL-97 program and refined with SHELXL-2014 by full-matrix least-squares on |F|2 values. The CCDC data of complex **2** is 1903857.

### 3.2. Synthesis of Complex **2**

A mixture of $Co(NCS)_2$ (0.42 mmol) and $H_2dpPzda$ (0.38 mmol) in methanol (50 mL) was stirred overnight and then the solution was filtered to remove impurities and concentrated under vacuum. Natural volatilization obtained deep orange single crystals (33% yield). EA (%) $C_{18}H_{20}CoN_8O_2S_2$: calcd. C 42.94, H 3.97, N 22.27, S 12.72; found: C 43.42, H 3.61, N 22.39, S 12.68; UV-Vis ($CH_3OH$) $\lambda$max/nm ($\varepsilon$/dm$^3$mol$^{-1}$·cm$^{-1}$): 206 ($1.70 \times 10^5$), 208 ($2.48 \times 10^5$), 360 ($1.96 \times 10^5$); IR(KBr, cm$^{-1}$): 3416 (s); 2050 (w); 1635 (s); 1385 (m); 1151 (m); 772 (w); 624 (w).

### 3.3. Catalytic Procedure

The detailed catalytic procedure was described in our previous work [33]. The catalyst, cocatalyst, and epoxide were sequentially added to a 100 mL high pressure reactor, then the reactor was pressurized to a required pressure with $CO_2$, and the mixture was heated and stirred for a while. After completion of the reaction, unreacted $CO_2$ was evacuated to remove pressure, the lid was removed, the product was taken out and separated, and the conversion of the epoxide was analyzed by $^1$H NMR spectroscopy.

## 4. Conclusions

In summary, we have successfully synthesized a novel cobalt complex $[Co(H_2dpPzda)(NCS)_2]\cdot CH_3OH$ (**2**) with $H_2dpPzda$ (**1**) as the chelating and bridging ligand. The structure of complex **2** is Co(II) coordinating with six nitrogen atoms to form an elongated octahedral geometry, with the pyrazine ring acting as a bridging ligand to link molecular units of complex **2** into a one-dimensional chain complex.

The complex **2** has been used for $CO_2$ conversion, and the results have shown that complex **2**/TBAB exhibits high catalytic activity for converting $CO_2$ into cyclic carbonates under mild conditions. The catalyst could be recycled to maintain the integrity of the structure and remained equal to the level of its catalytic activity even after 6 catalytic rounds, indicating that the novel 1 D supramolecular chain catalysts act as relatively robust materials.

**Supplementary Materials:** The following are available online at http://www.mdpi.com/2073-4344/9/11/951/s1, Table S1–S3: Crystal data of Complex **2**, Table S4: Highest Occupied Molecular Orbital (HOMO), and Lowest Unoccupied Molecular Orbital (LUMO) of complex **2**, Table S5: Representative homogeneous and heterogeneous cobalt catalysts with high TOF used for the synthesis of cyclic carbonates, Figure S1–S5: NMR spectra of carbon dioxide fixation reactions, X-ray molecular structure reported for complex **2**. (CIF).

**Author Contributions:** Data curation, W.F.; Formal analysis, W.F., L.W. and X.-G.J.; Funding acquisition, W.-Z.W., L.-L.L. and X.-G.J.; Investigation, W.F.; Methodology, W.F., L.W. and L.-L.L.; Project administration, W.-Z.W.; Writing—original draft, W.F.; Writing—review & editing, W.-Z.W., T.-C.X. and P.P.E.

**Funding:** This research was funded by the National Natural Science Foundation of China (No. 11847140), the Natural Science Foundation of Shannxi Province (No. 2019JZ-44; 2017JQ2009), the Natural Science Basic Research Plan in Shaanxi Province of China (No. 2019JQ-490), the Scientific Research Program Funds by Shannxi Provincial Education Department (No. Z18165), the Xi'an Shiyou University Postgraduate Innovation and Practical Ability Training Project (No.YCS17211017).

**Conflicts of Interest:** The authors declare no conflicts of interest.

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
