# Peer review of "Novel Cobalt Complex as an Efficient Catalyst for Converting CO2 into Cyclic Carbonates under Mild Conditions"

_catalysts, doi:10.3390/catal9110951_

Round 1

Reviewer 1 Report

The present manuscript, due to Wang and co-workers, describes the ability of a novel Co2+ complex to catalyze the insertion reaction of CO2 with epoxides to render cyclic carbonates. It operates with a significantly low catalyst loading and relatively mild conditions. Its activity is comparable to that of other previously described systems, although it does't imply a big improvement.

The work is concise and the experiments are well supported. I recommend this work for publication if the following points are addressed:

-Terming complex 2 as supramolecular is rather misleading (i.e. title, line 50, line 197), if not incorrect. The supramolecular properties of complex 2, if any, are not described or analyzed in the manuscript. It is better referring it as a metal-organic coordination polymer. Please note that Reference 14, an authoritative and comprehensive review, doesn’t term this sort of complexes as supramolecular assemblies. Please correct it.

-In the introduction section, line 34, references (reviews preferably) of CO2 insertions mediated by organocatalysts should be included. Some of these organocatalysts are very active and efficient and deserve attention, moreover considering that they’re “greener” than metal-based catalysts.

-Line 49, a reference for the ligand should be included at this point (reference 21).

-Line 172, Reference 20, is wrong. It doesn’t describe the preparation of ligand 1. Please remove it and replace it by Reference 21.

-Line 192, a fully detailed experimental catalytic procedure should be included again.

Reviewer 2 Report

This manuscript describes the cobalt complex-catalyzed synthesis of cyclic carbonates from CO2 and epoxides. Among many catalysts reported for the cycloaddition, the catalytic system showed a remarkable TOF and broad substrate scope of epoxide. The catalyst was easily recycled and reused. The structure of catalyst was well studied and a possible catalytic mechanism was proposed. From these reason, I conclude this work is suitable for publication in Catalysts. Before publication, please consider the following points.
1) P1 line 26-27: The sentence is too much to say because authors method required relatively high temperature and pressure. Room temperature and atmospheric pressure conditions would not discharge of CO2.
2) Scheme 2 line 160: Is it possible to indicate CO2 polarized by Pyridine as N-C=OO(-)?
3) Can the ring opened bromohydrin be obtained when an epoxide was treated with equimolar catalyst? If so, Scheme 2 is plausible. And if bromohydrin can not be obtained effectively, the ring opened process may be reversible.

Reviewer 3 Report

The authors report one-dimensional metal complex with the nitrogen polydentate ligand H2dpPzda and application to catalytic reaction of epoxides oxide with CO2 to give the corresponding carbonates. The Co complex as a catalyst was found to be a unique one-dimensional polymer structure. In addition, the catalytic reaction using a metal polymer complex proceeded under a mild condition. I recommend its publication in Catalysis after the below comments being taken into consideration.

In page 4, Table 1, please provide the blank experiment without the use of catalysts and co-catalyst. In addition, entry 16 showed high TOF at higher temperature of 120 deg. In this condition, the product may be formed by only co-catalyst TBAB as entry 3. Please provide the catalytic reaction without use of 2 as a blank experiment. This may be important to show an advantage of 2 in the catalytic reaction. In page 1, lines 34 and 38, please provide some references for “and the like”. In page 3, line 102, please provide references of previous reports of catalytic reactions for the sentence “The result show … Co-Salen and so on”. How about move Table S5 in SI to the main text. In page 5, Table 2, please modify structural formula of “Product”.
